# Tumor Microenvironment in Melanoma Brain Metastasis: A New Potential Target?

**DOI:** 10.3390/ijms26115018

**Published:** 2025-05-23

**Authors:** Gerardo Caruso, Cristofer Gonzalo Garcia Moreira, Edvige Iaboni, Massimo Tripodo, Rosamaria Ferrarotto, Rosaria Viola Abbritti, Luana Conte, Maria Caffo

**Affiliations:** 1Unit of Neurosurgery, Department of Biomedical and Dental Sciences and Morphofunctional Imaging, University of Messina, 98125 Messina, Italy; cristofer.garciamoreira@studenti.unime.it (C.G.G.M.); edvige.iaboni@studenti.unime.it (E.I.); massimo.tripodo@studenti.unime.it (M.T.); rosamaria.ferrarotto@studenti.unime.it (R.F.); maria.caffo@unime.it (M.C.); 2Service de Neurochirurgie, Hopital Laribosiere, 75010 Paris, France; rosaria.abbritti@aphp.fr; 3Department of Physics and Chemistry, University of Palermo, 90128 Palermo, Italy; luana.conte@unipa.it

**Keywords:** brain metastases, circulating tumor cells, melanoma, microenvironment, vascular mimicry

## Abstract

Melanoma, a malignant skin tumor, is the third skin tumor and the third cause of brain metastases. The development and introduction of systemic therapies, such as Braf inhibitors and checkpoint inhibitors, have guaranteed an increase in overall survival. The appearance of brain metastases, which determines a median survival of less than 5 months, represents a sign of systemic disease progression and tumor instability. In this view, in addition to systemic therapy, the therapeutic options can be surgery, stereotactic surgery, and whole-brain radiation therapy. However, it has been observed that the response to systemic therapies of brain metastatic lesions, compared to extracerebral ones, does not guarantee complete local tumor control, thus increasing the mortality and morbidity of patients. This phenomenon, tumor escape, makes systemic therapy partly ineffective. How melanoma cells migrate, cross the blood–brain barrier, and invade brain tissue is still being studied. The melanocytic metastatic brain tumor microenvironment and its assay seem to have a key role in the response and therefore in the progression of metastatic lesions. Through this work, the intent is to underline the importance of the brain tumor microenvironment and how it can influence tumor growth, its response to therapy, and the patient’s overall survival.

## 1. Introduction

Melanoma is considered the most aggressive skin tumor and accounts for 90% of deaths related to skin cancers [1]. In a recent study, it was observed that the incidence of melanoma has risen sharply among individuals over 60, and rates are expected to keep increasing for decades to come [2]. Melanoma is a leading cause of metastases in the central nervous system (CNS), representing 6–12% of all metastatic brain tumors [3]. Between 40% and 75% of patients with stage IV melanoma develop brain metastases (BMs), highlighting the remarkable capacity of melanoma to colonize the brain [4]. Melanoma brain metastases (MBMs) are linked to a poor prognosis, directly causing death in 60–70% of melanoma patients. Magnetic resonance (MR) is the gold standard for detecting and assessing malignant brain tumors, including BMs (Figure 1) [5].

Various studies have identified several risk factors for the development of MBMs, including male sex, age over 60, melanomas originating on mucosal surfaces or the trunk, head, neck, or scalp, as well as deeply invasive or ulcerated primary tumors. Additional high-risk characteristics include acral, lentiginous, or nodular histology, involvement of three or more regional lymph nodes at diagnosis or recurrence, and visceral metastases [6]. Key prognostic factors in MBM patients include the number, size, and location of BMs, the presence and extent of extracranial metastases, lactate dehydrogenase levels, and broader indicators such as patient age, tumor burden, and overall performance status [5].

The development of BMs is a multistep process that begins with the circulating tumor cells (CTCs) from the primitive tumor. Melanoma cells exhibit high brain tropism due to the overexpression of specific receptors and proteins, including chemokine receptor type 4 (CCR4), tetraspanins, integrins, melanotransferrin, and S100A4 protein, all of which play a critical role in the formation of MBMs [6]. Signaling pathways like PI3K/AKT and MAPK, which regulate cell survival and proliferation, are also heavily implicated in MBMs. Activation of the PI3K/AKT pathway contributes to both intrinsic and acquired resistance to BRAF and MEK inhibitors.

Patients with symptomatic MBMs and/or leptomeningeal metastases face a poor prognosis, marked by rapid disease progression and neurological decline [2]. This aspect represents one of the main factors of morbidity and mortality, accompanied by a narrowing in the different therapeutic modalities that can be offered to the patient. In this sense, the need to guarantee a wider therapeutic range has meant that, in the last decade, some aspects that allow the development of cerebral metastasis from melanoma have been re-evaluated. The ability of CTCs to circulate in the vascular network, eluding the immune system, as well as the mechanisms to overcome the blood–brain barrier (BBB) and the subsequent ability to settle and therefore, of the tumor conditions and especially of the tumor microenvironment or “metastatic niche”, are under continuous magnifying light to identify which molecules are crucial in this complex mechanism of tumor diffusion. Recognizing molecules and/or molecular pathways in the tumor microenvironment would allow us to offer more specific therapies to slow down CTCs, limit the formation of cerebral metastatic lesions, and therefore be able to guarantee an increase in overall survival by reducing morbidity and mortality.

## 2. Current Melanoma Brain Metastases Therapies

At present, there are three principal strategies to manage MBMs: surgery, radiation therapy (whole-brain radiotherapy (WBRT), stereotactic radiosurgery (SRS)), and systemic therapies (including protein kinase inhibitors (BRAF) and immune checkpoint inhibitors (cytotoxic T-cell antigen 4 (CTLA-4) or programmed cell death protein 1 (PD-1)). With a deeper understanding of the biology and the advancement of effective BRAF-targeted therapies and immunotherapy—alongside significant progress in local treatments like stereotactic radiosurgery—the prognosis for MBMs continues to improve, with the 1-year overall survival rate approaching 85% [6].

The advantages of surgery include rapid relief of the pressure effect on surrounding normal brain structures and evacuation of intra- or peri-tumoral hemorrhage, as well as procurement of tissue for diagnosis and molecular studies to select appropriate systemic therapy [7].

The role of WBRT is shrinking mainly due to its detrimental effect on neurocognition and quality of life. It is reserved for patients with diffuse brain involvement [8]. Nevertheless, the prognosis is dismal, with the median overall survival following WBRT approximately 2–5 months, and 1-year survival less than 10–12% [9,10]. The WBRT continues to play an important role in palliative care in patients who have numerous symptomatic brain metastases, extensive metastasis not amenable to radiosurgery, and symptomatic leptomeningeal carcinomatosis, and for those whose performance status is poor [6]. In selective patients with 1–4 metastases measuring less than 3–4 cm, SRS yields an excellent local control with response rates of as high as 90%, median survival of 5–11 months, and 1-year survival of 25% [11]. Generally, SRS can be used as an alternative to surgery in patients with lesion sizes of 3 cm each, and deep tumor locations, in those who are unable to undergo surgery [6].

Cytotoxic chemotherapy in MBMs has largely been ineffective without evidence of survival benefit [8]. Dacarbazine has long been the historical standard cytotoxic chemotherapy for melanoma to incite an appropriate response for the management of MBMs. Temozolomide, a derivative of dacarbazine, which readily penetrates the BBB to achieve cerebrospinal fluid concentration up to 30% of plasma, only demonstrated clinical responses in roughly 10% as a single agent or combined with WBRT or other agents [12].

Genetic mapping of melanoma has revealed that approximately 50% of cases harbor a driver mutation in the BRAF gene, a serine-threonine protein kinase, leading to uncontrolled activation of the MAPK pathway [13]. Currently, three FDA-approved BRAF inhibitors—vemurafenib, dabrafenib, encorafenib—are used in combination with the MEK inhibitor rametinib—cobimetinib, binimetinib, binimetinib—for metastatic melanoma treatment [13]. While BRAF inhibitors demonstrate strong single-agent activity with an initially promising response, resistance is almost inevitable. This resistance typically arises from inadequate MAPK inhibition or reactivation of the pathway through various molecular mechanisms, such as MEK mutations, BRAF-splice variants, NRAS mutations, or adaptive RTK signaling. As a result, combining BRAF and MEK inhibitors has become a key therapeutic strategy, aiming to concurrently target the PI3K-AKT and MAPK pathways. For patients with MBMs, the COMBI-MB trial found that dabrafenib–trametinib combination therapy achieved a higher intracranial response rate (58%) compared to single-agent BRAF therapy in patients with asymptomatic BRAF V600E-mutated MBM without prior local treatment [14,15].

Melanoma is a highly immunogenic cancer, which led to high-dose interleukin-2 (IL-2) being the first immune therapy used for stage 4 melanoma patients. However, IL-2 treatment was associated with severe toxicity and showed limited efficacy against brain metastases. The second FDA-approved immune therapy, interferon-α (IFN-α), also yielded limited responses. The landscape of immunotherapy changed with the approval of checkpoint inhibitors, including ipilimumab, a human monoclonal antibody that blocks CTLA-4, and nivolumab and pembrolizumab, which block PD-1 to stimulate antitumor T cell responses. Although immune therapies generally have poor penetration through the intact BBB, studies have shown that activated T cells can cross the BBB, suggesting that therapies stimulating T cell responses may be effective against MBMs [16]. Current evidence from phase II trials indicates that combination immunotherapy with ipilimumab and nivolumab provides a durable intracranial response in patients with asymptomatic MBMs.

## 3. The Microenvironment in Melanoma Brain Metastases

The remarkable aggressiveness of melanoma is due to its high genetic instability, which allows it to evade the responses of the immune system and conventional therapy [17,18]. The extensive and widespread colonization of melanoma cells affect various organs, such as the liver, bones, spine, and CNS. The metastatic phase of a tumor has two main protagonists: the CTCs and the tissue microenvironment, the peripheral site of the tumor that, following the acquisition of structural and vascular factors, ensures the settlement and growth of tumor cells. The interaction between these two axes is responsible for the diffusion, infiltration, and secondary settlement of CTCs. In this sense, in melanoma, the CTCs, through a chronic inflammation framework, would determine structural modifications in the CNS, thus creating an environmental setting suitable for their development [19]. The CNS is separated, and at the same time protected, by the BBB, a dynamic structure consisting of a complex cellular system composed of endothelial cells with their tight-junction adhesions, pericytes, and astrocytes with their extensions surrounded by a basement membrane [20,21]. The interaction that would be determined between CTCs and the BBB would be the basis of the accomplishment of CTCs and their subsequent migration (Figure 2).

The BBB allows the selective passage of molecules thanks to tight junctions and adherens junctions, and to the presence of low pinocytosis activity, low levels of leukocyte adhesion molecules, lack of fenestrae, and efflux pump expression [21]. During the tumorigenesis process, the dynamism of the BBB allows modifications at both cellular and molecular levels, giving rise to a new vascular-nervous unit, called blood–tumor barrier (BTB). This new entity is characterized by a series of dilatations of vessels, bizarre distribution of pericytes, loss of astrocytic pedicles, and neuronal connections that are the basis of a non-uniform permeability. These events are associated with altered transport mechanisms, such as an increase in efflux transporters or ATP-binding cassette transporters (ABC transporters). In this way, the BTB becomes able to interact with the CTCs, making the adopted pharmacological therapies less effective [22,23].

Once the CTCs produced by the primary tumor enter the vascular and/or lymphatic system, they present all the characteristics necessary to guarantee the metastatic process. They share antigens with endothelial cells. CTCs express on their surface molecules such as MCAM/MUC18, L1-CAM, ALCAM, NCAM, N-cadherin, VCAM, ICAM, CEA-CAM, PECAM, and VE-cadherin, thus managing to survive and subsequently adhere to vascular cells for tissue invasion [24]. According to the study by Zigler et al., the MCAM/MUC18 molecular complex is highly expressed in the advanced stages of the primary tumor (73%) and in metastatic cells compared to (90%) normal melanocytic cells (i.e., in nevi, 50%), demonstrating how this molecular product is relevant for progression and diffusion [25]. For this last aspect, it has been seen that this molecular complex is involved, in addition to heterotypic cellular adhesion, in homotypic interaction, favoring the formation of emboli of metastatic melanoma cells, a way of escaping from immunosurveillance. CTCs produce a chronic systemic inflammatory state. This inflammatory state could be the condition sine qua non for a “permissive niche” to form so that these cells can perform a possible migration and cerebral extravasation through the BBB. This is how the so-called premetastic niche is formed [26,27]. Only a small fraction of that pool of CTCs will be able, due to the inability to adapt after arrest and extravasation into the brain parenchyma, to be the seed of possible tumor growth [26,27]. With the release of cytokines, chemokines, exosomes, and angiogenic factors, the BBB would become more susceptible to structural and cellular modifications, favoring brain tumor colonization. In this sense, Wang et al., in their study, highlight the possible role of exosomes, produced by circulating melanoma tumor cells with action at the level of the endothelial cells of the BBB and subsequent structural modification [28]. According to Peinado et al., this condition implies the preparation of an adequate parenchymal medium before their arrival [29]. In this sense, exosomes have a major role in remodeling the tumor microenvironment and the formation of the premetastatic niche, influencing CTCs. This phenomenon is defined as tumor exosome-driven education, which, according to Aliotta et al., can determine alterations in gene expression and signaling pathways [30]. In fact, it was seen through the study of Kuroda et al. that the CD46 protein would play an important role in the internalization of melanocytic tumor exosomes [31]. Peinado et al. in their paper, isolated circulating exosomes from the plasma of human patients affected by melanoma at different stages, identifying the presence of very late antigen 4 (VLA4), heat-shock protein 70 (HSP70; an HSP90 isoform), MET oncoprotein, and tyrosinase-related protein 2 (TYRP2), defining their composition [29]. They found that patients with stage IV melanoma with an exosomal protein concentration lower than 50 mgr/mL had longer survival than those with a protein concentration higher than 50 mgr/mL, as well as the expression of TYRP2 increased in individuals who develop metastasis [29]. In addition to the production of factors such as TNF-α, TGF-β, and VEGF, the production of metalloproteinases, such as MMP 2 and MMP 9, with the destruction of TJ, associated with the downregulation of ZO-1, claudin-5, and occludin would increase the permeability of the BBB [32]. As follows, melanoma CTCs, whose duration in circulation is about 14 days, a longer duration than other CTCs of other primary tumors, can initiate the process of brain invasion [33]. This is also favored, according to Soto et al., by the overexpression of cell adhesion molecules (CAMs) such as E-selectins, VCAM, and ICAM on the cerebral endothelial surface, whose ligands are expressed instead on the surface of CTCs, thus favoring a greater susceptibility to colonize the CNS [24].

As a result, metastatic melanoma cells metastasize in a predictable way. This phenomenon, according to Langley and Fidler, is defined as the “seed and soil hypothesis”, according to which tumor cells, the seed, find the appropriate microenvironment of the organ, the soil, in which to grow [34]. Indeed, intrinsic seed mechanisms, such as the EMT (epithelial-mesenchymal transition) program, the existence of tumor stem cells, autophagy, metastatic dormancy, and other intrinsic pathways associated with extrinsic seed factors, such as extracellular vesicles, exosomal microRNAs, cytokines, and chemokines, are at the basis of the remodeling of the primary microenvironment and the preparation of the secondary “soil” microenvironment [35]. In this sense, as highlighted by the study by Kushiro et al. [36], adipocytes, in an inflammatory context, through the release of IL-6, promote the increase in the expression of genes involved in EMT, such as Snai1, MMP9, Twist, and Vimentin, and the decrease in the expression of genes such as E-cadherin and Kiss1, thus favoring the invasion capacity of CTCs. Through these mechanisms, the seed induces the formation of the premetastic niche.

Once metastatic cells can cross the BBB, the next phase of survival begins. In this sense, it has been seen that the perivascular niche, metabolism, and local cells are components of the tumor microenvironment responsible for metastatic success. This success is guaranteed by the site of arrival, in the vicinity of the endothelial vascular structures, the so-called perivascular niche, where, through structural vascular remodeling and endothelial differentiation, a specific metastatic cell pool can grow. This cerebral microvascular reorganization is essential to guarantee metastatic needs because the cells would find themselves in a hypoxic and nutrient-deprived environment without these modifications. This can occur through two processes: vascular co-optation and a particular pattern of vascularization, vascular mimicry [37]. The interaction that is created between metastatic cells and pre-existing cerebral vessels without angiogenesis is defined as cell–cell interaction vascular co-optation, a phenomenon proposed as one of the hallmarks of metastatic initiation [38]. This represents the first step in brain colonization. In this way, metastatic cells, through integrins and molecules such as LCAM1, interact with the endothelial basal lamina, obtaining nutrients, oxygen, and factors produced by endothelial cells (angiocrine factors), which are essential in the early stages of brain colonization. When angiogenic development is inversely proportional to tumor growth, the so-called vascular mimicry occurs to ensure the subsequent phases and the maintenance of metastatic cells [39]. In conditions of hypoxia, there is an increase in the production of the HIF complex, which regulates oxygen homeostasis, both in physiological and pathological conditions. This complex, produced by metastatic cells, induces the expression of factors such as VEGF, VEGFR, VE-cadherin, EPHA2, Twist, and Notch. Specifically, the interaction between the HIF complex and Notch would promote the dedifferentiation of tumor cells. Thanks to this genetic plasticity, linked to high chromosomal instability (CIN), metastatic melanoma cells can acquire the profile of endothelial cells, join to form vascular channels, connect with surrounding endothelial vascular mosaics, and once stable, as a vascular structure, blood flow becomes regular. The hypoxic condition, with the production of HIF complexes, also includes mitochondria. In fact, as exposed by the study of Comito et al., hypoxia determines an increase in reactive oxygen species (ROS) that stabilize with the ability of hypoxia-inducible factor-1α (HIF-1α) to form capillary-like structures through vasculogenic mimicry [40]. The above requires that certain characteristics of the tumor microenvironment occur, such as hypoxia, which ensures an angioneurotic-like switch, which in turn determines an acidic environment, necessary for the remodeling of tumor cells, generated by mitochondria, through oxidative phosphorylation and increased ROS. Fisher et al., in their study, emphasize that BMs from melanoma preferentially use oxidative phosphorylation, suggesting such pathways as potential targeted therapies [41]. Such changes are not only guaranteed by circulating tumor cells but also by brain parenchymal cells. Zou et al. have highlighted that PPARgamma pathways, which are created by the interaction between astrocytes and metastatic melanocytic cells, are important in glucose and lipid metabolism [42].

## 4. CTCs and Cells Around the Microenvironment

The maintenance of the microenvironment is not only associated with the interaction of CTCs with vessels and BBB, but also with the cells nearby at the borders of the tumoral niche, like astrocytes, tumor-associated macrophages (TAMs), and tumor-infiltrating lymphocytes (TILs).

Astrocytes, in addition to playing a role in brain tissue homeostasis, in the maintenance of the BBB, and in the process of neuroinflammation, interact with metastatic melanocytic cells [43]. Astrocytes could protect melanoma CTCs from chemotherapy, and so by tumor cell apoptosis, sequestering cytoplasm calcium through direct functional gap junctions via connexin 43, but their role in facilitating brain colonization is unknown [44]. Astrocytes communicate with each other by producing cytokines that, in addition to exerting a paracrine effect, also influence metastatic cells, improving migration and invasiveness in the latter. This interaction is reciprocal: metastatic melanoma cells induce the production, by astrocytes, of pro-inflammatory factors such as IL-23, and on the other hand, factors such as MMP2 are produced. Klein et al., in their study, highlight how IL-23, through the expansion and maintenance of IL-17, keeps a chronic inflammatory state active and at the same time determines an overregulation of MMP2, important in metastatic invasion [45]. This inflammatory picture also demonstrates the presence of the so-called tumor-infiltrating immune cells made up of T lymphocytes, macrophages of B lymphocytes, plasma cells, mast cells, and natural killer (NK) cells [46]. T lymphocytes are composed of various subpopulations or activation states: cytotoxic T lymphocytes (CD8 coreceptor) can kill, like NK, the neoplastic cells, and the CD4 helper T lymphocytes act indirectly via cytokine secretion. From the perspective of antineoplastic activity, TH1 lymphocytes are the most important because of their role in activating TC lymphocytes, and the CD4-expressing T-cell subpopulation also comprises regulatory T lymphocytes playing a role in antigen tolerance and suppression of effector T lymphocytes [47]. In the melanoma brain metastasis, dense infiltration of CD3+/CD8 T cells and low levels of CD4 and NK cells increased their infiltration in the brain parenchyma [26]. Lymphocytic infiltrates are mostly located in the tumor’s peripheral area and the surrounding brain: CD8+ lymphocytes are usually observed in perivascular areas, and CD4+ and Treg lymphocytes are dispersed intratumorally [48]. This peculiar distribution of TIL is correlated with the areas with intense angiogenesis. Moran et al. have found fewer vessels but larger ones in the peripheral area than in multiples with small diameters in the core of the microenvironment [49]. This underlines a different way of angiogenesis where new vessels emerge from the vascular remodeling of large vessels by a process called nonsprouting angiogenesis or vascular co-option [50]. This phenomenon, which reflects an increased mean vascular density at the periphery of a tumor, is correlated with poor prognosis.

Jacob et al. said high numbers of perivascular TREG lymphocytes with elevated expression of CTLA4 and FoxP3 have a high MIB1 index [51]. Lymphocytic infiltration was correlated with PD-L1 expression, graded prognostic assessment (CD3+ lymphocytes and edema (CD8+ lymphocytes) [52]. As well as tumor-associated macrophages (TAMs, also including microglia) are classified according to their activation status as dormant M1 (ramified) or activated M2 (amoeboid). The M1 macrophages are primarily proinflammatory and antineoplastic, with the ability to kill tumors, present antigens, and produce immunostimulatory cytokines. Instead, most TAMs belong to the M2 subclass, with reduced antigen-presenting ability, proangiogenic and pro-invasive phenotype, and secreting immunosuppressive cytokine [53]. Cancer cells synthesize colony-stimulating factor 1 (CSF1) to stimulate macrophages. This aspect causes the clustering of TAMs and cancer cells around blood vessels and promotes intravasation and the formation of metastases [54]. TAMs, so, might even proceed with the BM development as they were shown to be involved in the preparation of the premetastatic niche [55].

Even if in the melanoma brain metastasis scenario, there is a lack of evidence regarding the role of myeloid cells, these subtypes of cells play an important role in immunosurveillance. In the brain, it is recognized that central nervous system-native myeloid cells (CNS-myeloid) contribute to brain homeostasis and diseases, and peripheral bone marrow-derived myeloid cells (BMDM) infiltrate the brain parenchyma and contribute to neuroinflammation. CNS-myeloid and BMDM are implicated in brain metastasis progression [56]. Guldner et al., in their paper, highlight how CNS-myeloid cells, by CxCl10, facilitate a metastatic niche suppressing T-cells [57].

It is evident that the immunological aspect of the microenvironment is important in the success of melanoma metastasis. Like so, microglia, despite killing and phagocytizing tumor cells in the early stages of brain invasion, following the release of intrinsic factors, downregulate genes involved in phagocytic activity, such as Tmem119 or Trem2, switching from an anti-tumor phenotype to a pro-tumor phenotype [58]. Metastatic progression is associated with the activation of the NF-κB signaling pathway in these cells. Rodriguez-Baena et al. have shown that Rela/NF-κB depletion is associated with the reprogramming of microglia towards a pro-inflammatory phenotype, thus having a deleterious role in the metastatic microenvironment of the brain [59]. Fischer et al., in their study, highlighted how an immunosuppressed tumor microenvironment also depends on other factors, such as high levels of oxidative phosphorylation by inhibiting, for example, PGC1α, MITF, and mTOR and glutamine [60].

Therefore, it is easy to understand how the immune aspect, through its ability to create an environment rich in cytokines and pro-inflammatory factors responsible for pro-tumor immune cellular phenotypes, is important in metastatic maintenance and development.

## 5. Discussion

One of the biggest problems faced by melanoma patients is brain metastases, which occur in up to 60% of patients with advanced melanoma. The continuous progress of conventional therapies, such as chemotherapy and radiotherapy, in primary tumors has allowed for local control of tumor growth, guaranteeing the patient an increase in long-term survival. This growth of therapeutic response is often lacking in the presence of metastases, especially at the cerebral level, both due to the presence of the BBB, which limits the arrival of chemotherapy drugs, and because the molecular cascades underlying metastatic development are still unclear, thus proving not completely adequate in tumor control in advanced stages. Usually, BMs are more permeable than healthy brain tissue. At the same time, permeability is not homogeneous across the entire brain tumor barrier, influencing the degree of permeability. The study by Lyle et al. highlights how the cellular and molecular architecture of the BTB is altered compared to the BBB. The study highlights how the degree of permeability depends on the type of pericyte subpopulations and laminin α2 of the parenchymal basement membrane [61].

The treatment of cerebral metastases remains, therefore, currently controversial both for the indications, based on size, number, location, and the Karnofsky performance scale (KPS), and for the type of treatment, conventional or surgical. The intrinsic malignancy of the metastatic lesion, the toxicity of chemotherapeutics, the neurological and neurotoxic effects related to radiotherapy, and the morbidity and mortality associated with the surgical procedure have brought the importance of interrupting those mechanisms that make metastatic cells “fit” for brain colonization to the center of research. From the above, the process of metastatic formation of melanoma is made up of several steps that are crucial for the survival of CTCs. Therefore, the identification of molecules involved in the different phases of this phenomenon can be targeted to slow down, block, and prevent metastatic tumor growth (Table 1).

It is well established that anti-angiogenic drugs, such as VEGF inhibitors, play an important role in blocking tumor angiogenesis with negative effects on tumor and metastatic growth. In melanoma, the slow adaptation of the tumor microenvironment in developing new vessels creates a hypoxic environment that induces, due to a lack of nutrients and oxygen, metastatic cells to develop new vascular patterns such as vasculogenic mimicry. Due to the different molecular constitutions underlying this phenomenon, it has been observed that VEGF inhibitors have little efficacy in blocking VM and at the same time increase hypoxia in the metastatic microenvironment. Xu et al., in their study, highlighted that the use of bevacizumab, a VEGF-A inhibitor, in the short term would determine a reduction in tumor growth and an adaptive effect of metastatic cells known as metastatic conditioning [62]. Furthermore, in the same study, the cell culture media treated with bevacizumab, up to 48 h after inoculation of the drug, had not undergone any influence either on cell viability or on the formation of VM, determining instead an increase in the latter. Liu et al., in their study, both in vitro and in vivo, showed that apatinib (known as YN968D1, a new agent for anti-angiogenic therapy) could inhibit the expression of VEGFR-2, and downregulate the ERK1/2/PI3K/MMP-2 signaling cascade involved in VM formation, thus overcoming the negative effects caused by blocking angiogenesis and at the same time reducing VM [63]. Norcantharidine, which, according to the results obtained from the study by Wang et al., inhibits tumor growth and VM formation of melanoma suppressing matrix metalloproteinase-2 expression [64]. As well as curcumin, imatinib, and thalidomide have all been shown to inhibit melanoma VM, concomitant with decreases in EPHA2, VE-cadherin, PI3K, VEGF, HIF-1, MMP-2, and MMP-9 expression and/or activity. As explained above, chronic inflammation represents the platform where the tumor microenvironment, following modifications and overexpression of protein complexes, becomes suitable to receive CTCs. This dynamic interaction is the basis of metastatic tumor progression. In this scenario, a significant role is played by the melanoma cell-adhesion molecule or MCAM, one of the main molecules involved in homotypic cell adhesion, i.e., between metastatic melanocytic cells, and heterotypic cell adhesion with endothelial cells to ensure microvascular adhesion and endo- and trans-endothelial migration. Specifically, this occurs, as explained by Wang et al., from the interaction between MCAM and S100A8/A9 [64]. MCAM is overexpressed on the surface of CTCs, playing an important role in the metastatic progression of melanoma; it can also be used to identify/quantify CTCs and therefore be a predictor of treatment response [65]. S100A8/A9, a heterodimeric complex present both intracellularly and extracellularly especially in neutrophils and monocytes, by binding to some proteins such as activated leukocyte cell adhesion molecule (ALCAM), melanoma cell adhesion molecule (MCAM) and receptor for advanced glycation end products (RAGE) which are tumor grade dependent, intervenes in the growth, adhesion and degree of invasiveness of CTCs [64]. Once the S100A8/A9-MCAM complex is formed, two main pathways are activated: NF-kB, which regulates proliferation, survival, and tumor migration, further increasing the expression of MCAM, and the production of ROS, which fuels the pre-existing inflammatory state and instead increases the expression of S100A8/A9 [65]. Furthermore, NF-kB would seem to be involved in the mechanisms of chemoresistance of melanocytic cells through the repair of damaged DNA [65]. S100A8/A9, in addition to being expressed at the pulmonary level, is widely expressed, under high levels of chronic inflammation, in the brain too [64]. A plasma glycoprotein, produced by the liver, histidine-rich glycoprotein (HRG) that could efficiently regulate the activity of extracellular S100A8/A9, preventing the development of organotrophic metastases, both cerebral and pulmonary, mediated by S100A8/A9 [66]. At the brain level, through exosomes derived from primitive melanocytic cells, microglia and astrocytes produce S100A8/A9, increasing the inflammatory loop and making the microenvironment suitable for metastatic development. Tomonobu et al., in their study, found that low levels of HRG are inversely proportional to melanocytic metastatic growth both at the brain and lung levels [67]. However, if the multitude of molecular entities and signaling pathways regulating the proliferation and cellular survival/cell death is considered, the inhibition of a singular target gene or transcriptional factor could not be sufficient to suppress neoplastic progression. The development of a metastasis is a complex and multifactorial mechanism involving a large series of molecules and cell–cell interactions. These processes allow individual tumor cells to migrate and invade the brain parenchyma. Proteases, cell adhesion molecules, and their related signaling pathways show an important role in neoplastic cell migration and invasion. The complexity and crosstalk between signal transduction pathways limit the potential efficacy of targeting a single receptor or molecule. Furthermore, many patients develop resistance to BRAF inhibitors with mechanisms that are still poorly understood. Recently, in addition to mechanisms related to genomic or epigenetic abnormalities, the tumor microenvironment also seems to play a role in the mechanisms of resistance to BRAF inhibitors. The hyperactivity of cancer-associated fibroblasts (CAFs) and tumor-associated macrophages (TAMs) appear to promote resistance of neoplastic cells to bRAF inhibitors [68].

Mitochondria and oxidative phosphorylation play an important role in maintaining the tumor microenvironment. Oxidative phosphorylation, as mentioned above, appears to play a role in inhibiting the immune response through the production of glutamine, resulting in shorter survival. IACS-010759 [69] has been shown to inhibit mTOR signaling as well as CB839 [70], a novel small molecule glutaminase inhibitor, achieving a low oxidative phosphorylation profile and immunoenhancement, opening broader and combined therapeutic fields.

Hence, it is observed that angiogenesis and the chronic inflammatory state, hallmarks of tumor development, which are at the basis of the formation of the microenvironment, are not static but highly dynamic phenomena where the different methods that primitive and metastatic tumor cells use to survive, migrate, bypass the immune response, overcome physiological barriers and form their own niche for protection and subsequent tumor transformation, ensure that each tumor transformation is cell- and organ-specific. This is to underline that the analysis of the tumor microenvironment in melanoma brain metastases requires further studies aimed at identifying target molecules and pathways to “overcome” the obstacles that arise with the current therapy, making it more suitable and specific to ensure better tumor surveillance in terms of overall and progression-free survival.

The tumor microenvironment or niche represents a protected environment within which the tumor cell can grow, thus creating, through the mechanisms mentioned above, a substrate for tumor proliferation, peritumoral inhibition, and intratumor resistance. The tumor microenvironment plays a key role in the control of neoplastic progression and in the response to medical therapies. The search for new targets and related mechanisms may stimulate the investigation of new therapeutic protocols. Therefore, it seems plausible to consider that the tumor microenvironment and the mechanisms that regulate it may represent potential therapeutic targets in the treatment of melanoma brain metastases.

## Figures and Tables

**Figure 1 ijms-26-05018-f001:**
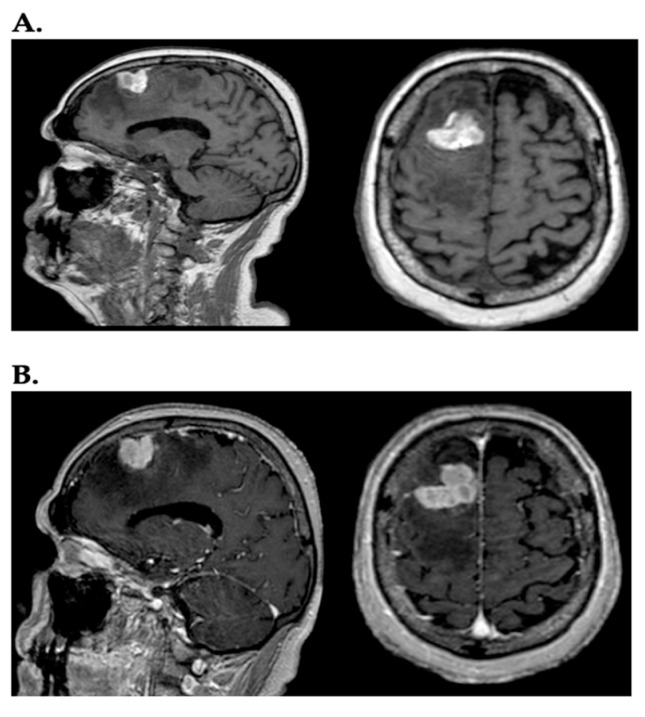
(**A**) In the sagittal and axial planes, a lesion, heterogeneously hyperintense, with irregularly oval morphology and diffuse perilesional edema, is visible in the right frontal region on T1-weighted images; (**B**) the same lesion after administration of contrast (gadolinium) assumes intense contrast enhancement.

**Figure 2 ijms-26-05018-f002:**
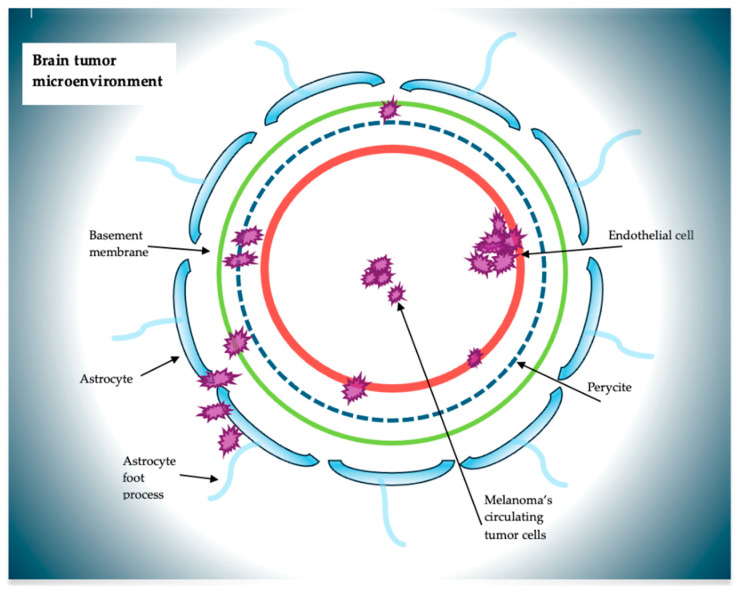
Schematic representation of the main agents and their interactions within the brain tumor microenvironment.

**Table 1 ijms-26-05018-t001:** Principal molecular targets and their antagonist involved in the development of melanoma brain metastasis and their tumoral microenvironment.

Principal Target Therapies	Molecular Targets and Activities
Apatinib (YN 968N1)	Block VEGFR-2 reducing vascular mimicry
Norcantharidine	Inhibits MM2, vascular mimicry decreasing and tumor colonization.
Histidine-Rich Glycoprotein (HRG)	Block S100 A8/A9, reducing inflammatory loop making microenvironment not suitable for metastatic development
IACS-010759	Block M-TOR, reducing oxidative phosphorylation, reducing glutamine concentration, increase immunocomponents
CB825	Decreasing of glumatine, increase immunocomponents

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
