# Peer review of "Tumor Microenvironment in Melanoma Brain Metastasis: A New Potential Target?"

_ijms, 2025, doi:10.3390/ijms26115018_

Round 1

Reviewer 1 Report

Comments and Suggestions for Authors

Dear authors,

The manuscript provides a solid and comprehensive narrative overview of the cellular and molecular mechanisms involved in MBMs, while highlighting promising potential therapeutic targets.

Strengths of the manuscript:

  • The topic is of high relevance to the oncology and neuro-oncology communities.
  • The manuscript is well structured, moving logically from the clinical background to molecular pathophysiology and emerging therapeutic strategies.
  • The review is richly documented, with a substantial number of up-to-date references, reflecting solid background research.
  • The authors provide a thorough overview of the TME, including astrocytes, tumor-associated macrophages, infiltrating lymphocytes, exosomes, and vascular mimicry mechanisms.
  • Promising molecular targets such as the S100A8/A9-MCAM-NF-κB axis are discussed in a meaningful and clinically relevant manner.

Minor revisions recommended:

  1. Language and Style: The manuscript would greatly benefit from professional English editing by a native speaker to improve clarity, grammar, and syntax.
  2. Redundancy: Some sections (e.g., on circulating tumor cells or exosomes) include partially redundant information. These could be streamlined and shortened for better readability.
  3. Discussion Section:
    • The discussion could be strengthened by adding a more critical perspective on the literature, including the limitations of certain therapeutic strategies (e.g., anti-angiogenic therapies and resistance to BRAF/MEK inhibitors).
    • Translational perspectives and potential clinical applications of targeting the microenvironment should be more explicitly discussed.
  4. Optional (but encouraged):
    • A summary table of key molecular targets and associated therapeutic strategies in the TME of MBMs.
    • A second table summarizing recent or ongoing preclinical or clinical studies targeting the brain metastatic microenvironment.
  5. References: While numerous and generally relevant, the integration of clinical trials (e.g., COMBI-MB, CheckMate-204) into the critical analysis could be enhanced.

Conclusion:

This is a well-constructed and highly relevant review that offers a valuable synthesis of current knowledge on the role of the tumor microenvironment in melanoma brain metastases. It is suitable for publication in IJMS pending minor revisions, mostly related to language polishing and light improvements to the critical and translational depth of the discussion.

Comments on the Quality of English Language

The manuscript would greatly benefit from professional English editing by a native speaker to improve clarity, grammar, and syntax.

Author Response

RE: “Tumor Microenvironment in Melanoma Brain Metastasis: A New Potential Target?”

Manuscript ID: ijms-3623042

Reviewer #1

We thank the Reviewer for your suggestions. We have carefully read the reviewer’s comments and have revised our manuscript accordingly.

Language and Style:

  • The manuscript would greatly benefit from professional English editing by a native speaker to improve clarity, grammar, and syntax.
  • As suggested by the reviewer, we have checked the text for spelling, grammatical and syntactic errors.

Redundancy:

  • Some sections (e.g., on circulating tumor cells or exosomes) include partially redundant information. These could be streamlined and shortened for better readability.
  • In accordance with the reviewer's suggestion, we have removed redundant data in the revised text.

Discussion Section:

  • The discussion could be strengthened by adding a more critical perspective on the literature, including the limitations of certain therapeutic strategies (e.g., anti-angiogenic therapies and resistance to BRAF/MEK inhibitors).
  • In agreement with the reviewer, we have modified the discussion by adding the following concepts: “However, if the multitude of molecular entities and signaling pathways regulating the proliferation and cellular survival/cell death is considered, the inhibition of a singular target gene or transcriptional factor could not be sufficient to suppress neoplastic progression. The development of a metastasis is a complex and multifactorial mechanism involving a large series of molecules and cell-cell interactions. These processes allow individual tumor cells to migrate and invade the brain parenchyma. Proteases, cell adhesion molecules and their related signaling pathways show an important role in neoplastic cell migration and The complexity and crosstalk between signal transduction pathways limit the potential efficacy of targeting a single receptor or molecule. Furthermore, many patients develop resistance to BRAF inhibitors with mechanisms that are still poorly understood. Recently, in addition to mechanisms related to genomic or epigenetic abnormalities, the tumor microenvironment also seems to play a role in the mechanisms of resistance to BRAF inhibitors. The hyperactivity of cancer-associated fibroblasts (CAFs) and tumor-associated macrophages (TAMs) appears to promote resistance of neoplastic cells to BRAF inhibitors [68].

  • Translational perspectives and potential clinical applications of targeting the microenvironment should be more explicitly discussed.
  • In agreement with the reviewer's suggestion, we have modified the final part of the discussion: “The tumor microenvironment or niche represents a protected environment within which the tumor cell can grow, thus creating, through the mechanisms mentioned above, a substrate for tumor proliferation, peritumoral inhibition, and intratumor resistance. The tumor microenvironment plays a key role in the control of neoplastic progression and in the response to medical therapies. The search for new targets and related mechanisms may stimulate the search for new therapeutic protocols. Therefore, it seems plausible to consider that the tumor microenvironment and the mechanisms that regulate it may represent potential therapeutic targets in the treatment of melanoma brain metastases.”

  • A summary table of key molecular targets and associated therapeutic strategies in the TME of MBMs.
  • In accordance with the review, we have added Table 1 to the text.

Table 1: Principal molecular targets and their antagonist involved in the development of melanoma brain metastasis and their tumoral microenvironment.

Principal Target Therapies

 Molecular targets and activities

Apatinib (YN 968N1)

Block VEGFR-2 reducing vascular mimicry

Norcantharidine

Inhibits MM2, vascular mimicry decreasing and tumor colonization.

Histidine-Rich Glycoprotein (HRG)

Block S100 A8/A9, reducing inflammatory loop making microenvironment not suitable for metastatic development

IACS-010759

Block M-TOR, reducing oxidative phosphorylation, reducing glutamine concentration, increase immunocomponents

CB825

Decreasing of glumatin, increase immunocomponents

  • References: While numerous and generally relevant, the integration of clinical trials (e.g., COMBI-MB, CheckMate-204) into the critical analysis could be enhanced.
  • We have changed the references.

Reviewer 2 Report

Comments and Suggestions for Authors

This review addresses an important and timely topic: the role of the brain tumor microenvironment in melanoma brain metastases (MBMs). The manuscript offers a reasonable overview but would benefit from deeper biological insights. 

Blood-Brain Barrier (BBB) Disruption
The authors state that several therapies, as reflected in the following sentence — “Dacarbazine has long been historical standard cytotoxic chemotherapy for melanoma, but it's not able to cross the BBB to incite appropriate response for the management of MBM” — or in the paragraph “Although immune therapies generally have poor penetration through the intact BBB, studies have shown that activated T cells can cross the BBB, suggesting that therapies stimulating T cell responses may be effective against MBMs,” assume that the BBB remains intact. However, it should be noted that, in general, macrometastases exhibit a glomeruloid vascular architecture characterized by increased fenestration and elevated hydrostatic pressure, which also contribute to the reduced efficacy of chemotherapy.

Seed and Soil Hypothesis
The concept is briefly mentioned but deserves a more in-depth discussion. A detailed analysis linking melanoma cell-specific characteristics (“seed”) with brain-specific microenvironmental factors (“soil”) would greatly enhance the understanding of cerebral tropism.

Superficial Analysis of the Tumor Microenvironment
The discussion of immune cells (TILs, TAMs, astrocytes) is relevant but superficial. The review should delve further into how an immunosuppressive niche is established, the metabolic adaptations involved (e.g., preferential use of oxidative phosphorylation in MBMs), and the role of astrocyte–melanoma cell interactions.

Imbalance Between Therapy and Biology
The therapeutic sections (e.g., surgery, radiotherapy, systemic treatments) are disproportionately long compared to the discussion of the metastatic biology, thereby diluting the biological focus of the review.

Other Comments
Several sentences are confusing or poorly constructed (e.g., “the hypoxic condition, with the production of HIF complexes, also includes mitochondria”), and require revision to improve clarity and scientific precision.

Author Response

RE: “Tumor Microenvironment in Melanoma Brain Metastasis: A New Potential Target?”

Manuscript ID: ijms-3623042

Reviewer #2

We thank the Reviewer for your suggestions. We have carefully read the reviewer’s comments and have revised our manuscript accordingly.

Blood-Brain Barrier (BBB) Disruption

  • The authors state that several therapies, as reflected in the following sentence — “Dacarbazine has long been historical standard cytotoxic chemotherapy for melanoma, but it's not able to cross the BBB to incite appropriate response for the management of MBM” — or in the paragraph “Although immune therapies generally have poor penetration through the intact BBB, studies have shown that activated T cells can cross the BBB, suggesting that therapies stimulating T cell responses may be effective against MBMs,” assume that the BBB remains intact. However, it should be noted that, in general, macrometastases exhibit a glomeruloid vascular architecture characterized by increased fenestration and elevated hydrostatic pressure, which also contribute to the reduced efficacy of chemotherapy.
  • In agreement with the reviewer, we have consequently modified the text in the “The Microenvironment in Melanoma Brain Metastases” section: “During the tumorigenesis process, the dynamism of BBB allows modifications at both cellular and molecular levels giving rise to a new vascular-nervous unit, called blood-tumor-barrier (BTB). This new entity is characterized by a series of dilatation of vessels, bizarre distribution of pericytes, loss of astrocytic pedicles and neuronal connections that are the basis of a non-uniform permeability. These events are associated with altered transport mechanisms, such as an increase in efflux transporters or ATP-binding cassette transporters (ABC transporters). In this way, the BTB becomes able to interact with the CTCs, making the adopted pharmacological therapies less effective [22-23]; and in the “Discussion” section: Usually, BMs are more permeable than healthy brain tissue. At the same time, permeability is not homogeneous across the entire brain tumor barrier, influencing the degree of permeability. The study by Lyle et al., highlights how the cellular and molecular architecture of the BTB is altered compared to the BBB. The study highlights how the degree of permeability depends on the type of pericyte subpopulations and laminin α2 of the parenchymal basement membrane [61].

Seed and Soil Hypothesis

  • The concept is briefly mentioned but deserves a more in-depth discussion. A detailed analysis linking melanoma cell-specific characteristics (“seed”) with brain-specific microenvironmental factors (“soil”) would greatly enhance the understanding of cerebral tropism.
  • In agreement with the reviewer we have modified the text: "Indeed, intrinsic seed mechanisms, such as the EMT (epithelial-mesenchymal transition) program, the existence of tumor stem cells, autophagy, metastatic dormancy and other intrinsic pathways associated with extrinsic seed factors, such as extracellular vesicles, exosomal microRNAs, cytokines and chemokines are at the basis of the remodeling of the primary microenvironment and the preparation of the secondary “soil” microenvironment [35]. In this sense, as highlighted by the study by Kushiro et al., [36] adipocytes, in an inflammatory context, through the release of IL-6, promote the increase in the expression of genes involved in EMT, such as Snai1, MMP9, Twist, and Vimentin, and the decrease in the expression of genes such as E-cadherin and Kiss1, thus favoring the invasion capacity of CTCs. Through these mechanisms, the seed induces the formation of the premetastic niche.”

Superficial Analysis of the Tumor Microenvironment

  • The discussion of immune cells (TILs, TAMs, astrocytes) is relevant but superficial. The review should delve further into how an immunosuppressive niche is established, the metabolic adaptations involved (e.g., preferential use of oxidative phosphorylation in MBMs), and the role of astrocyte–melanoma cell interactions.
  • It is evident that the immunological aspect of the microenvironment is important in the success of melanoma metastasis. In this sense, microglia, despite killing and phagocytizing tumor cells in the early stages of brain invasion, following the release of intrinsic factors, downregulate genes involved in phagocytic activity, such as Tmem119 or Trem2, switching from an anti-tumor phenotype to a pro-tumor phenotype [58]. Metastatic progression is associated with the activation of the NF-κB signaling pathway in these cells. Rodriguez-Baena et al., have shown that Rela/NF-κB depletion is associated with the reprogramming of microglia towards a pro-inflammatory phenotype, thus having a deleterious role in the metastatic microenvironment of the brain [59]. Fischer et al., in their study, highlighted how an immunosuppressed tumor microenvironment also depends on other factors, such as high levels of oxidative phosphorylation by inhibiting, for example, PGC1α, MITF, and mTOR and glutamin [60].

Imbalance Between Therapy and Biology

  • The therapeutic sections (e.g., surgery, radiotherapy, systemic treatments) are disproportionately long compared to the discussion of the metastatic biology, thereby diluting the biological focus of the review.
  • In agreement with the reviewer, we have shortened the specific sections.

Other Comments

  • Several sentences are confusing or poorly constructed (e.g., “the hypoxic condition, with the production of HIF complexes, also includes mitochondria”), and require revision to improve clarity and scientific precision.
  • We have checked the text and corrected any inaccuracies.